# Automatic and Intelligent Technologies of Solid-State Fermentation Process of Baijiu Production: Applications, Challenges, and Prospects

**DOI:** 10.3390/foods10030680

**Published:** 2021-03-23

**Authors:** Hong Ye, Juan Wang, Jie Shi, Jingyi Du, Yuanhao Zhou, Mingquan Huang, Baoguo Sun

**Affiliations:** 1Key Laboratory of Brewing Molecular Engineering of China Light Industry, Beijing Technology and Business University, Beijing 100048, China; yehcn@163.com (H.Y.); btbuwangjuan@163.com (J.W.); shijie2968@163.com (J.S.); 17888820115@163.com (J.D.); zhouyuanhao97@163.com (Y.Z.); 2Beijing Laboratory of Food Quality and Safety, Beijing Technology and Business University, Beijing 100048, China

**Keywords:** Baijiu, automatic technology, intelligent technology, solid-state fermentation, solid-state distillation

## Abstract

Baijiu is the national liquor of China and the world’s most consumed spirit, which is produced using a unique and traditional solid-state fermentation (SSF) process. The development of an automatic and intelligent technology for SSF is more difficult than that for liquid-state fermentation. However, the technological upgrading of the SSF process is crucial for reducing the labor intensity, saving manpower, avoiding the waste of materials and energy, and providing a favorable operation environment for workers; moreover, it provides a great reference value to similar industries. This article reviews the latest application progresses in automatic and intelligent technologies for Baijiu production. The important technical processes are introduced successively, including the production of Jiuqu, SSF, solid-state distillation, storage, and blending. The bottlenecks and challenges are pointed out for automatic and intelligent upgrading of these technical processes. Furthermore, the typical technology application cases in an integrated automatic production line of Baijiu are also summarized. Next, the industrial development status of Baijiu production is compared with those of other liquors in the world. Finally, future development directions are proposed. This review will provide an overall introduction and objective analysis of recent developments and current challenges in Baijiu manufacturing so as to promote the intelligent brewing of Baijiu.

## 1. Introduction

Baijiu is the national liquor of China; with a 2000-year history, it is one of the oldest distilled liquors in the world. Baijiu is the world’s most-consumed spirit; for example, the gross sales of Baijiu in 2019 were more than $80 billion [1]. Furthermore, it is very famous not only in China but also in other countries worldwide [2]. Baijiu, brandy, whisky, vodka, gin, and rum are considered as six of the most famous distilled liquors manufactured worldwide [3]. According to the flavor characteristics, Baijiu can be classified into 12 flavor styles: Nong- (strong), mild (light), Jiang- (soy sauce), Mi-, Feng-, Chi-, Te-, herbal, Fuyu-, sesame, Laobaigan-, and Jian- (Nongjiang-) flavor types. The Nong-flavor type is the most important representative of the Baijiu flavors followed by the mild flavor. As the production processes shown in Figure 1, the raw materials are mixed with whole or powdered grains and are then wetted and cooked. Then, the mixture in a solid state is cooled to the appropriate temperature (about 18–35 °C), mixed with Jiuqu (Koji), and placed in a fermentation jar or pit. When the fermentation is finished, the fermented grains remaining in the solid state are removed for distillation to obtain the base liquor. After a long storage period followed by blending, the commercial liquor can be obtained.

As shown in Figure 1, it represents a traditional solid-state fermentation (SSF) process that all of the materials during transportation, mixing, fermentation, distillation, and reuse are solid. Almost all of the famous high-quality Baijiu brands in China are produced using SSF, except for the Mi- and Chi-flavor Baijiu. In contrast, other distilled liquors worldwide are generally the products of liquid-state fermentation (LSF).

SSF is more challenging than LSF for the following reasons. First, from the perspective of material transportation, the efficiency of the conveying equipment for solid materials is not as high as that of a pump for liquid transportation. Second, from the perspective of heat transfer, the heat exchange among solid materials is more difficult than in liquids. Thus, the temperature distribution in solid materials is often not uniform, making the product quality unstable. This is especially evident during the processes of Daqu production and SSF. Third, because the required temperature variation for SSF is slow and moderate, the traditional fermentation containers (such as pits or earthenware jars) are usually set underground. As such, it is difficult to monitor their fermentation parameters, resulting in a “black box” fermentation process. Fourth, most of the processes in the production of Baijiu (e.g., fermentation, distillation, and storage) are batch operations, in which the materials require manual operations and a long residence time. As a result, it is difficult to apply unified modern food processing equipment and automatic production lines to the manufacture of Baijiu. Nevertheless, during the last decade, there have been great developments in the automation and intellectualization of Baijiu manufacturing. It is a pity that few reports have focused on the advances in this area.

In this paper, recent advances in the automatic and intelligent manufacturing of Baijiu are reviewed comprehensively and critically in order to provide an overall introduction and objective analysis of Baijiu manufacturing for international and domestic engineers and other relevant persons. It also has important reference value for the manufacturing of other distilled liquors and similar products produced using SSF.

## 2. Production of Jiuqu

Jiuqu is a type of fermentation starter; it is China-specific, and it includes Daqu, Xiaoqu, and Fuqu. Daqu is the oldest starter in China and is widely used for the production of Nong-, mild, and Jiang-flavor Baijiu. The raw materials of Daqu mainly include wheat, barley, and peas, which are processed into a block with a size of 25 cm × 15 cm × 5 cm and weight of 3–4 kg. The main raw materials of Xiaoqu are early indica rice and wheat; they are cultivated under controlled temperature and humidity, after inoculation using ripened Jiuqu. Sometimes, Xiaoqu is in a granular form and does not need to be shaped, whereas other times, it is processed into spherical particles with sizes smaller than those for Daqu. Xiaoqu is usually used to produce Huangjiu, as well as mild, Chi- and Mi-flavor Baijiu. The microorganisms in Daqu and Xiaoqu from the air or the surroundings contain molds, yeasts, bacteria, and actinomycetes. The molds and yeasts are similar in Daqu and Xiaoqu, and *Rhizopus*, *Aspergillus*, *Saccharomyces cerevisiae,* and *Pichia* are the dominant microbes. However, there are great differences in bacterial community members. The dominant bacteria in Xiaoqu are *Lactobacillus*, *Weisseria*, *Pediococcus*, *Klebsiella*, *Acinetobacter,* and *Acetobacter*, while the dominant bacteria in traditional Daqu are *Bacillus*, *Enterobacter,* and so on [4]. Fuqu is mainly made from wheat bran, and it requires artificial inoculation with a pure mold. According to the different requirements for Fuqu, the main microbes used to make Fuqu are *Rhizopus*, *Aspergillus niger*, *Aspergillus oryzae*, *Aspergillus Albicans*, *Saccharomyces cerevisiae, Saccharomycopsis fibuligera, Enterococcus faecium, Clostridium beijerinckii, Bacillus cereus,* and *Acetic acid bacteria* sp. [4]. Fuqu usually does not need to be processed into shapes, and it can be applied for the production of almost all flavor styles of Baijiu and Huangjiu. However, Fuqu is seldom used alone in the production of famous Baijiu brands with high quality.

Among three kinds of Jiuqu, Xiaoqu, and Fuqu often do not need to be shaped in large-scale production. Furthermore, their production periods are short. Thus, mechanized and automatic production was achieved relatively early. For example, Kuaijishan Shaoxing wine Co., Ltd. (Shaoxing, China) pioneered automatic control in the production of raw wheat Qu (one kind of Xiaoqu) and built an automatic production line with an annual output of 4000 tons of raw wheat Qu. This was an important transformation in the making of Jiuqu, i.e., from traditional manual to automatic production. Soon afterwards, in 2013, Jingpai Co., Ltd. (Huangshi, China) achieved the automatic production of mixed Xiaoqu, using a disk starter propagation machine as the main equipment. The development of automation in Daqu-making has been relatively slow due to its process characteristics.

The Daqu preparation process can be divided into two steps: the preparation of Daqu blocks and cultivation of Daqu. The traditional operations of the former step such as shaping by manual trampling have been replaced by mechanization equipment such as crushing, blending, and molding machines. Men et al. [5] suggested that the different parts of the former step could be linked by a belt conveyor or chain and pan conveyor in the production of Daqu. When a programmable logic controller (PLC)-based automatic control system, human–machine interactions, and a frequency converter are incorporated in the actuator system, the automatic control system is formed and can be realized in the feeding and crushing of the raw materials for the forming of Daqu blocks. This would be helpful in obtaining Daqu with stable quality. At present, there are still two concerns regarding the production of Daqu: (1) providing real-time monitoring of the parameters and automatic operations for turning over Daqu in starter-culturing workshops (the latter step); and (2) building a complete set of continuous and automatic Daqu production lines to combine the shaping processes with the cultivation processes.

The automatic carrying and turning over of Daqu in starter-culturing workshops is the foremost problem to be solved. The temperature in a workshop for Daqu-culturing can be as high as approximately 40–60 °C [6]; under such adverse working conditions, workers can only keep working for less than half an hour. Furthermore, Daqu is much heavier than Xiaoqu, and therefore, the carrying and turning over of Daqu requires a higher labor intensity. During the traditional process of Daqu-culturing in the workshop, the manual operations and observations require frequent entries and exits of workers, resulting in low production efficiency and the possible introduction of unexpected bacteria. Thus, there is an urgent need for an automatic Daqu incubation operation. In the last decade, devices for the automatic turnover of Daqu have been reported [7,8,9]. Daqu can be placed on shelves and turned over by mechanical and automatic devices. In addition, owing to the popularity of the Internet of Things, the real-time monitoring of incubation parameters in the workshop and observation of the apparent morphology of Daqu during incubation are possible. The apparent morphology of Daqu is mainly observed through human eyes after enlarging the images or pictures, including the mycelial growth state of mold, which could be reflected by the appearance of some white filaments on surfaces of Daqu, as well as the color changes on sections that reflect the growth state of other microorganisms. This lays the foundation for building a digital and intelligent Daqu-making workshop.

In 2012, Anhui Gujing Group Co., Ltd. (Bozhou, China) built a digital comprehensive management and control system for a Daqu-culturing workshop based on the Internet of Things [10]. The system mainly includes three parts. The first part is a sensing layer comprising three wireless sensors for the real-time monitoring of the temperature (intra and between Jiuqu), humidity, and concentration of carbon dioxide (CO_2_). The problems regarding the sensitivity and accuracy of these sensors have gradually been solved. In the second part, the data transporting layer, the signals collected by these sensors are transmitted to a local area network of the company for data sharing. Thus, each department can access the required data, so as to remotely monitor the process and dispose the need for mobile devices. The third part comprises the design of the practical application. Workers can quickly inspect all starter-culturing workshops online to acquire the apparent morphology of the Daqu. If necessary, the Daqu can be turned over by an automatic device [11], as shown in Figure 2. As a result, the quality of the Daqu is stable and uniform. The application of a digital comprehensive management and control system for starter-culturing workshops greatly reduces the labor intensity, improves the working environment, and stabilizes and increases the quality of the Daqu. For example, the inspection workload for the workshop is decreased by three to five times, and the engineers obtain more detailed and complete digital information from the Daqu incubation process. In 2017, Sanjing Liquor Industry Co., Ltd. (Botou, China) applied a digital control system based on the wireless sensing technology and the Internet of Things in the production of Shilixiang Daqu; this decreased the average labor cost by 150,000 CNY for a workshop with an output of 2000 tons of Daqu per year [12].

The realization of digital and intelligent monitoring enables the establishment of an automatic production line for Daqu. In recent years, the automatic production of Daqu has been realized. For example, in 2019, Sichuan Jiannanchun Group Co., Ltd. (Mianzhu, China) applied an intelligent Koji-making system for the production of Daqu for Nong-flavor Baijiu (NFB) [13], as shown in Figure 3. The system provided automatic feeding, crushing, weighing, mixing, adjustment of the mixed water amount, quick detection of moisture, shaping of the Daqu, and intelligent management of the incubating bacteria. The microbial diversity indicators and dominant microbial community in the intelligently made Koji are consistent with those in traditional artificial Koji. For example, according to the research reported by Jiang et al. [14], the number of bacteria (10^7^), molds (10^5^), and yeasts (10^3^–10^4^) were similar in the mechanically made and traditional artificial Jiang-flavor Daqu. Additionally, a total of 84 bacterial genera were detected in mechanically made Daqu by Zuo et al. [15], which was slightly higher than that in traditional koji (82 bacterial genera), and the dominant bacterial genera were *Pantoea*, *Rhizobium*, *Lactobacillus*, *Weissella*, *Bacillus*, *Oceanobacillus*, *Lentibacillus*, *Kroppenstedtia*, *Thermoactinomyces*, *Staphylococcus*, *Enterobacteriaceae*, *Saccharopolyspora*, *Leuconostoc*, and *Pseudomonas*. Furthermore, the physiochemical property indicators are slightly better than those in artificial Koji. In the Daqu starter-making workshop, the control system could intelligently adjust the temperature and humidity of the workshop according to “big data”, greatly improving the quality of the Daqu and brewing abilities. For example, the standard deviation of the weight of artificial Koji is 0.430, whereas that of intelligently made Koji is 0.115. In addition, the automatic shaping system can produce 800–1000 pieces of Daqu per hour; each Daqu-shaping machine is equivalent to a labor load of 15–20 workers. Thus, significant labor costs are saved.

## 3. Solid-State Fermentation

SSF is a unique process in traditional Baijiu brewing, and it represents the greatest difference from the other famous distilled spirits. During SSF, the saccharification, fermentation, and subsequent distillation are conducted using solid materials with a solid content of approximately 60%. Although the liquor yield per weight of the grains in SSF is lower than that in liquid fermentation, the quality of the SSF liquor is evidently better than that of LSF liquor, forming a typical feature of Chinese liquor.

SSF is a complex exothermic process involving microbial activity. Furthermore, thermal convection is difficult for the air in the gaps between solid grains. As a result, parameters such as the fermentation temperature are heterogeneous in time and space. In addition, the mechanized transportation of fermented grains is difficult, owing to the limitations of the underground pit or jar. The fermentation process determines the type and quantity of aroma components in the Baijiu; thus, the effective monitoring of the fermentation process is of great significance for ensuring the stability of the Baijiu quality. Recent concerns regarding SSF mainly include the monitoring of fermentation parameters, innovation in fermentation containers, and developing equipment for the continuous transportation of materials in or out of the underground fermentation pit.

### 3.1. Monitoring of Fermentation Temperature

There are three ways to monitor the temperature of SSF: manual measurement, wired temperature measurement, and wireless temperature measurement.

The manual temperature measurement requires workers to record the values of thermometers on site to acquire the temperatures at different depths in the pits. Evidently, manual measurement results in low accuracy and rough positioning. The wired temperature measurement requires a prior positional arrangement for setting wires and sensors in the fermentation pits; this is costly and requires consideration of the aging of the wires under the fermentation environment. Furthermore, it is not practical to set up a large data monitoring center near the pits in a fermentation workshop. Therefore, wireless monitoring is an ideal way to monitor the fermentation parameters.

Zigbee technology is a short-range two-way wireless communication technology, and it is often used in wireless transmission in the Internet of Things [16]. Shandong Gubeichun Group Co., Ltd. (Dezhou, China) [17] and Jiangsu Yanghe Distillery Co., Ltd. (Suqian, China) [18] built intelligent pit monitoring systems based on Zigbee technology. The values of temperature measurements from different depths in the pits were sent to a local area network server through a Zigbee communication module. This reduced the subjective errors during manual measurement and enhanced the accuracy of the measurement. However, Zigbee technology suffers from high costs and short communication distances, and it is considered as unsuitable for industrial production [19]. Thus, the CC1101 wireless transmission module, with an open transmission distance of 300–500 m, has been applied for wireless measurements of temperature within an entire SSF workshop [19]. The effective detection of the three-dimensional temperature field in the pit provides reliable data for optimizing the SSF technology to improve the quality and yield of Baijiu.

The temperature during the fermentation process is not constant and generally must follow certain variations: the temperature slowly increases to a sufficiently high peak value, remains for some time, and then decreases similarly slowly. This variation trend of the fermentation temperature can benefit the formation of flavor components, but it is characterized by evident uncertainty, time variability, and nonlinearity. From the perspective of cybernetics, it comprises a control problem with uncertain complex processes. Liu [20] indicated that traditional control methods, such as neural network control, expert systems, and fuzzy logic control, were not appropriate for an SSF process. However, a human-simulated intelligent control strategy could imitate the control behaviors of experts, i.e., closer to the true situation of SSF. The simulation results showed that the method for temperature optimization control was reasonable and feasible. Lu et al. [21] collected data from an SSF pit to form a big data system. After analyzing and processing the big data, the optimal fermentation process was determined by an intelligent expert system based on cloud computing. However, its decision-making required long-term optimization training to achieve the optimal result.

The above studies not only propose new methods for the sensing and monitoring of temperature during an SSF process but also provide references for the intelligent control and optimization of pit temperatures. However, the automatic control of pit temperature has not been achieved in practice, as it is difficult to install temperature control equipment underground. In addition, the temperature varies with the depth of the pit; as such, the temperature should be controlled hierarchically. Therefore, although digital monitoring of the fermentation temperature in pits is not very difficult, the control of pit temperature is difficult (unless the location of the fermentation containers is changed).

### 3.2. Fermentation Containers

The traditional fermentation containers for Baijiu are jars or pits underground. The transportation of the grains depends on totally manual operation or a bridge crane (called “Hangche” in China) with manual auxiliary operation, which requires high labor intensity. In addition, to realize convenient monitoring and control of the fermentation process, the fermentation container must be changed not only from “underground” to “overground” but also from “fixed” to “mobile”.

Overground fermentation has been attempted since the 1990s. A tunneling fermentation chamber was designed, including mobile fermentation tanks and temperature-controlling equipment [22]. However, the effects on the quality of Baijiu were not reported. Yue et al. [23] designed an artificial pit (250 L of charge) for the fermentation processes of traditional Chinese liquor. As shown in Figure 4a, heating and cooling equipment using circulation water controlled the temperature of the mobile pit. The gas collection equipment was used for gas analyses and providing an air balance for the entire system. As shown in Figure 4b, the vent on the bottom of the pit was advantageous for the sampling and draining of Huangshui, and the design of the pulley was convenient for moving. Although the fermentation time was only 1 month, the yield ratio of the liquor from the artificial pit was 20.5%, compared with the typical 3-month fermentation yield of 33% in the industry. However, as compared with traditional fermentation, there was a nearly 30% decrease in the aromatic components (such as ethyl butyrate and ethyl hexanoate) in the liquor. It can be seen that the contradiction between the innovation of the fermentation container and retention of liquor flavor is the key problem in pit upgrading. In addition, temperature control based on circulation water in jackets is difficult to achieve in industrial production; air conditioning for temperature control is more practical.

At present, NFB is highly dependent on pit mud; the mud of the pit bottom and seal mud of the pit top are also of great importance for the production of Jiang-flavor Baijiu (JFB). This is because the flavors of these two liquors would be seriously influenced without the mud pit for both NFB and JFB. Mild flavor Baijiu (MFB) has relatively little dependence on pit mud, and underground jars are adopted as fermentation containers in traditional production. Furthermore, there is no requirement to remove the fermented grains from the pit layer-by-layer, as in the production processes of NFB and JFB. Thus, it is easier to design an innovative fermentation container for MFB than for NFB and JFB.

In 2010, Fenjiu Group Co., Ltd. (Fenyang, China) designed a rolling stirred fermentation tank, as inspired by a cement mixer for construction engineering [24]. The design of the rolling structure improved the full mixing of the materials. However, this design is still in the theoretical stage. Since 2010, the effects of fermentation containers on the fermentation process have been studied. Ma et al. [25] from Luzhou Laojiao Co., Ltd. (Luzhou, China) studied the effects of different fermentation containers. The results showed that stainless steel jars and pottery jars were more suitable for the fermentation of Daqu MFB than stone pits under the unique geographic conditions and climate in Luzhou. The stainless steel jars and pottery jars had similar liquor yields (stainless steel jars, 18.8%; pottery jars, 18.9%), and their liquors had the similar contents of total acids (stainless steel jars, 1.19 g/L; pottery jars, 1.08 g/L) and total esters (stainless steel jars, 3.25 g/L; pottery jars, 3.29 g/L). Comparatively, stainless steel containers have more significance for practical application than pottery jars, at least from the perspective of operation and production efficiencies. Stainless steel containers can be set underground or overground. Anhui Golden Seed Winery Co., Ltd. (Fuyang, China) [26] observed that the fermentation temperature variation in an underground stainless-steel pit was similar to that of an underground brick pit during the production of Xiaoqu MFB. They speculated that the underground soil environment (instead of the pit material) was the main factor influencing the fermentation temperature. Through pressure cooking, low-temperature fermentation, extending the fermentation cycle, and other new fermentation technology, the liquor yield in a stainless steel pit was 2–4% higher than in the brick pit. Furthermore, the content of fuel oil decreased by 20–30%, and the content of ethyl acetate increased by approximately 50%. For example, the liquor yield increased from 52.8% (the brick pit) to 55.5% (the stainless steel pit), the content of total esters, total acids, and ethyl acetate were also increased from 0.59, 0.31, and 0.448 g/L (the brick pit) to 0.82, 0.42, and 0.794 g/L (the stainless steel pit), respectively. The effect of the position of the fermentation container was also investigated [27]. The fermentation temperature in a stainless steel overground container demonstrated a slow increase and decrease within a small fluctuating range of 5 °C, owing to the good heat conductivity of the stainless-steel material. However, the temperature in the stainless underground container demonstrated an increase of approximately 10 °C after the grains were placed into the container for 3–7 days. On the ninth day of fermentation, the temperature began to decrease slowly. Although the above two containers showed evident differences in the variation of fermentation temperature, the contents of the total ester (4.44 g/L), ethyl acetate (2.12 g/L), and ethyl lactate (3.20 g/L) in the Baijiu produced in the overground stainless-steel vessel were slightly higher than those produced in the underground stainless-steel vessel (3.14, 0.86, and 2.18 g/L, respectively), except that the contents of total acids (1.11 g/L) in them were the same. It is more important that the application of the stainless-steel fermentation containers benefits the mechanized discharging of the fermented grains, so as to significantly reduce the labor intensity and facilitate the monitoring and control of the SSF.

In addition to the materials and position of the fermentation container, the volume of the fermentation container also has evident effects on the fermentation. Fenglin Distillery of Jingpai Co., Ltd. (Huangshi, China) used two types of stainless-steel fermentation tanks with volumes of 1 m^3^ and 2 m^3^, respectively [28]. The ethyl acetate content in the 1 m3 fermentation tank was approximately 1.5 times that in the 2 m^3^ tank, and the ratio of ethyl acetate to ethyl lactate was approximately 2.0. The fuel oil contents were similar for both tanks. Thus, reconstructing the 2 m^3^ large tank into two 1 m^3^ small tanks could increase the content of ethyl acetate and improve the quality of the Xiaoqu Baijiu.

The application of a stainless-steel fermentation tank (SSFT) is helpful for the continuous production of Baijiu. The feeding of the grains into the pit can be realized by a screw conveyor, and the discharging of the grains out of the tank can be realized using a rack to turn over the SSFT. In addition, the tanks can be carried by forklift, saving significant amounts of time and labor. However, there are still concerns regarding the use of SSFT. The first concern is the material stability of the SSFT in a fermentation environment. Under fermentation conditions, slow corrosion by acids may occur in the stainless steel, eventually leading to leakages. In addition, some heavy metal ions of nickel and chromium may be released from the stainless steel to the grains and liquor. Another concern is the fact that SSFT currently cannot be used for the fermentation of NFB and JFB, because SSFT cannot provide pit mud. Thus, equipment that can provide a more appropriate environment for SSF must be developed [29].

### 3.3. Continuous and Automatic Transportation for Grains in or Out of Pit Underground

Based on the use of an SSFT, the feeding and discharging of grains has become easier than before for the production of MFB (but not NFB and JFB). The fermentation processes for NFB and JFB remain underground, and continuous and automatic operation remain difficult.

In a traditional transportation mode, the grains are put into and taken out of the underground pit via being grasped by a bridge crane, as assisted with manual operation. For example, the crane can mechanically place the grains into the pit; then, workers must manually pave them smooth. After fermentation, the fermented grains are removed from the pit in the same manner (i.e., using a bridge crane with manual assistance). This operation is evidently intermittent and inefficient. Moreover, the space of the mud pit is so narrow that the mud on the wall and bottom of the pit is easily damaged by grabbing actions from the crane, even with relatively little shaking of grab. In addition, the fermented grains must be transported layer by layer for the production of NFB and JFB, as the flavors of the liquors from fermented grains at different pit depths are quite different. It is difficult to remove accurately fermented grains layer by layer from an underground pit by grabbing. As a result, manual operation is currently necessary for the transportation of grains from underground pits.

In spite of this, the equipment has been designed for continuous fermented grain-discharging devices recently, including (1) a continuous grabbing device [30], as shown in Figure 5a, comprises a translation component, rotation component, quick-change component, and fermented grain-grabbing operating component; and (2) a continuous lifting device, with a structure similar to that of the scraper conveyer [31]. As shown in Figure 5b, the scraper continuously scoops the fermented grains and lifts them out of the pit. This device was applied by Anhui Jinzhongzi Distillery Co., Ltd. (Fuyang, China). Using the continuous lifting device, only two workers were required to complete the transportation of fermented grains out of one pit within 0.5 h, whereas the manual operation to discharge the fermented grains from one pit required three workers working for 1.5 h. Thus, the production efficiency was increased threefold [31]. In addition, a novel device combining a scraper conveyer and screw conveyor has been introduced [32]. During operation, the screw conveyor is used to collect grains to the inlet of the scraper conveyer; then, the grains are fed into the scraper conveyer and are carried away. The combined screw conveyor shortens the stroke of the scraper conveyer. The above-mentioned devices greatly improve the production efficiency and reduce labor costs. However, these continuous transportation devices are still not fully automatic, especially regarding the discharging of fermented grains from the underground pit, and they still require manual auxiliary operation. For underground fermentation processes, such as in the production processes of NFB and JFB, there is still a long way to go before achieving continuous automatic production.

## 4. Solid-State Distillation

Regardless of how the fermented grains are discharged from the fermentation containers, they must be distilled to obtain liquor (base Baijiu). The generation of flavor substances depends on fermentation process, whereas the extraction of them depends on distillation process. Both fermentation and distillation are extremely important for the production of Baijiu. The solid-state distillation (SSD) of fermented grains is an unsteady distillation process, and it uses grains (fermented grains or a mixture of fermented grains and fresh grains) as packing. When water vapor passes through the grains from bottom to top, the liquor in the fermented grains is vaporized into the water vapor, which is condensed to obtain the base liquor. A steaming bucket with a height of approximately 1 m is called a “Zeng” in China, which has an upper diameter of 1.7–2.0 m and bottom diameter of 1.6–1.8 m. The temperature and composition of the vapor change with the distillation time and axial direction of the Zeng. After decades of development, the traditional wooden Zeng has generally been replaced by a stainless-steel Zeng, which can be turned over (or has a separable bottom for conveniently discharging the distilled grains). For SSD, the following problems remain to be solved.

(1) SSD is a typical batch process, and it is difficult to achieve continuous production.

(2) The efficiency when extracting the flavor substances from the grains is very sensitive to the packing. An uneven packing of the grains will result in a channel flow, which will allow the vapor to quickly pass through a short channel with a lower resistance rather than through the gaps among the grains, i.e., for the full extraction of the aroma substances.

(3) The steamer-filling operation is difficult to be replaced using machines. This is because during the traditional steamer-filling process, the grains are dropped little-by-little by workers, for a gradual building of the grain column in a Zeng with vapor charging. In this process, a loose channel can be observed at some positions, i.e., where more vapor escapes than in other places. At this position, more grains are needed to press the leaking point. The steamer-filling operation is complicated, and it strongly depends on the subjective judgment and proficiency of the workers. Furthermore, this operation has a heavy workload.

(4) The mechanism of SSD has not been studied systematically, resulting in a practical operation that lacks theoretical guidance. For example, it is difficult to determine the exact end time point of distillation, or to achieve an accurate cutting of the fractions for such an unstable SSD process. The picking of the Baijiu is rough and experimental. The personal judgments of workers often result in large differences among the qualities of different batches of Baijiu.

To solve the above problems, a continuous SSD process and automatic and intelligent process for steam-filling and Baijiu picking have been developed.

### 4.1. Continuous Solid-State Distillation

Xu et al. [33] designed a continuous distillation system for Baijiu production by SSD; this is almost the only report regarding continuous distillation equipment for SSD. The fermented grains were fed and dropped from a hopper to the chain of a conveyor, and they were separated by a certain number of scrapers to form a series of small “movable steamer buckets”. The water vapor extracted the liquor when penetrating the conveyor belt (and the grain layer thereon). The vapor containing liquor was collected by a trap hopper above the conveyor belt, and then it was condensed into liquor. The distillation time could be adjusted by changing the speed of the conveyor belt. When the grains moved to the end of the belt, they dropped from the belt to a discharging outlet.

In the continuous distillation system, the fermented grains were formed as a thin layer horizontally instead of one column formed by gradual and vertical packing, and the distillation process was changed from “stationary” to “mobile”. As a result, it was possible to achieve automatic and continuous distillation. Both the production capacity and stability of the product were improved, and the liquor yield was increased by 1% [34]. Furthermore, the system could be driven by solar photovoltaic power, which is of great significance for realizing clean production, energy saving, and emissions reduction [35,36]. In 2013, based on the application of continuous distillation equipment, Jiangsu King’s Luck Brewery Joint-Stock Co., Ltd. (Huai’an, China) tentatively applied a computer control system to control the production of the liquor distillation process, so as to realize computer network management in the production process [37]. Based on the settings of parameters and feedback signals from sensors, such a system can control most parts of the production, such as cooking, feeding, distillation, liquor picking, and discharging. This system can benefit online detections of the liquor temperature, alcohol content, and so on. The liquor loss is reduced to 1%, owing to the operation under an enclosed space. In addition, the human cost is reduced because of the decreased labor intensity. However, such continuous distillation equipment does not seem to solve the problems regarding uniform grain layer construction, and the effects of such continuous distillation on the liquor quality have not been reported.

Until now, there have been very few reports on continuous distillation equipment for producing Baijiu by SSD. This may be owing to the unclear distillation mechanisms and complex and variable processes of SSD. Providing effective, applicable, and continuous distillation equipment will facilitate a significant revolution in the production of Baijiu.

### 4.2. Automatic and Intelligent Steamer-Filling

Although the batch operation of SSD is not easy to change, there has been great progress in automatic and intelligent steamer-filling, owing to rapid developments in automation technology (especially artificial intelligence technology).

Traditional steamer-filling involves the gradual accumulation of grains to build a uniform material column by dropping an appropriate amount of grains to press on the leaking point of water vapor, which requires a process of real-time detection. As shown in Figure 6, the process requires the careful observation and correct judgment of skilled workers, and the workers need to scatter a small amount of grains in a specific position quite frequently. Thus, the steamer-filling operation requires great labor strength, and workers often require several years of training and practicing for independent steamer-filling. Recently, the recruitment of workers for steamer-filling has become increasingly difficult, as most young people do not prefer work with such heavy labor intensity. Thus, steamer-filling urgently requires intelligent and automatic improvements.

Zhang et al. [38,39] designed an automatic distillation apparatus, “three-position” rotary table Zeng, for NFB produced by SSF. As shown in Figure 7, the “three-position” rotary table Zeng comprised three working positions on a rotary table, and different operations, including steamer-filling, distillation, and steamer-discharging, could be conducted simultaneously. When the rotary table turned by 120°, the steamer turned to the next position to start the next process. At each working position, the corresponding devices were equipped for specific operations. For example, a steamer-filling station was equipped with a screw conveyor and scattering device, and a steamer-discharging station was equipped with lifting and overturning devices. At the distillation position, a condenser and collector for liquor picking were installed. Notably, the steamer at the steamer-filling position could revolve on its axis, which improved the uniform distribution of the fermented grains. At present, a “four-position” rotary table Zeng has also appeared with one more position. This extra position is only used to extend cooking time after distillation [40]. Such semi-continuous operation systems have been applied in practical applications for Baijiu production, and the production efficiency has been improved, owing to the overlapping of operation times at different positions. One possible problem concerns the matching of the operation times at the three or four different positions. Another problem is that the manual assistance is still needed in this steamer-filling process [40].

With the rapid development of robot technology, robots for steamer-filling have begun to play important roles in the Baijiu brewing industry. Robots can provide intelligent and accurate judgments of working conditions, and they are not affected by the external environment. In 2015, Jiangsu King’s Luck Brewery Joint-Stock Co., Ltd. (Huai’an, China) developed a production line based on steamer-filling robots, thereby realizing a leap from traditional manual operation to intelligent operation in Baijiu manufacturing [41]. Robots for steamer-filling can solve the long-term problems caused by manual operation in Baijiu manufacturing and greatly improve the production efficiency.

At present, the robots for steamer-filling mainly include the following types:

(1) The steamer-filling robot that needs to fetch the fermented intermittently: This type of robot was used for steamer-filling in early applications. Based on an articulated mobile robot, the hopper is set on the ends of the robot for dropping the grains into the Zeng [42]. During the process of steamer-filling, the robot arm first moves to the feeding position to load the fermented grains, and then it moves to above the Zeng and begins scattering the fermented grains layer by layer. After finishing one round of filling the grains, the robot repeatedly loads and scatters the grains until the steamer-filling is complete. This type of robot, as assisted with infrared thermometry, was applied in Jiangsu King’s Luck Brewery Joint-Stock Co., Ltd. (Huai’an, China) and greatly increased the production efficiency, although the entire process was not continuous. Based on the application of this steamer-filling robot, a new automatic production line was built. The automatic production line achieved an 80% decrease in the amount of labor and a 10% increase in the rate of high-quality liquor relative to traditional production. Yang et al. [43] designed a LabVIEW-based automatic monitoring system for monitoring the vapor pressure and condensation temperature. This system could coordinate with a steamer-filling robot to achieve automatic and intelligent steamer-filling and distillation.

Another special type of bionic robot has been designed to imitate the movements and gestures of a human; this is quite different from the above-mentioned robots and is designed from the perspective of the motion of the robot arm. Sichuan Tuopai Shede Distillery Co., Ltd. (Suining, China) [44] used a six-degree-of-freedom robot to imitate the steamer-filling operations of the human body, mainly including the bending of the body and the action of the cranking arms. However, the machine could not adjust the operations according to feedback regarding the grain layer. In addition, the problem of the continuous transportation and filling of grains remained unsolved.

(2) The steamer-filling robot that can continuously transport and feed the grains: This type of robot has a mechanical arm by which grains can be continuously transported. The robot can also receive feedback from the grain layer to optimize the operation. For example, a robot vision system can be added to detect the thickness of the material layer [42]. Furthermore, the surface temperature of the fermented grains can be monitored in real time using infrared sensing technology, and the data from the detector can be transmitted to the robot so that it can adjust the positions where the grains are scattered [45]. A representative of this type of robot for steamer-filling made by Wuhan Fenjin Intelligent Machine Co., Ltd. (Wuhan, China) is shown in Figure 8, and was successfully applied at Fenglin Distillery of Jingpai Co., Ltd. (Huangshi, China). During the process of the steamer-filling by a robot, the grains are scattered continuously from the hopper into Zeng, as shown in the video (Appendix A). Depending on the feedback from the infrared thermometry, the robot arm can timely adjust the feeding position. Each steamer-filling robot is equivalent to a labor load of 3–4 workers. The operation of intelligent robots is 5–8 min faster than manual operation, and the liquor yield can be increased by over 4.34%, with an increase of 2% in the rate of first-grade products.

Some of the robots for steamer-filling being applied in the practical production of Baijiu are listed in Table 1. The improvements mainly include the reduction of labor intensity and steam consumption, which benefits the increase in production efficiency and saving of energy. Simultaneously, the yield of liquor with high quality also increases due to the decreased content of ethyl lactate and increased content of ethyl caproate.

After steamer-filling is finished, liquor picking is another problem that must be solved during the distillation process, because the liquors for different qualities should be collected separately.

### 4.3. Automatic and Intelligent Picking of Baijiu

The different Baijiu distillates throughout the distillation time have different components [49]. Thus, a separate collection of the distillates is necessary to classify the Baijiu into different products according to the ethanol content and flavor characteristics. A rough picking of Baijiu can be achieved by cutting the distillates into the foreshot of the liquor, the main body of the liquor, and the tail of the liquor, according to the ethanol content. However, it is difficult to complete a detailed classification and ranking according to the flavor characteristics during SSD. In general, the further classification of the Baijiu must be completed by specific tasters in the laboratory, and the process is time-consuming.

Currently, liquor picking based on ethanol content mainly depends on workers’ observations on the sizes of the so-called “Jiuhua”, i.e., the splashing foam or bubbles of liquor in the containers. As the surface tension of ethanol is less than that of water, the liquor (with a high ethanol content) forms large bubbles, which disappear quickly. When the ethanol content in the liquor decreases with the distillation process, the “Jiuhua” presents as smaller bubbles that disappear slowly. Depending on the workers’ judgment of the morphology of the “Jiuhua”, the distillates are separated into the three sections mentioned above for collection. Evidently, there is an overreliance on personal experience during the process of liquor picking, resulting in unstable quality. Therefore, there is an urgent need to develop intelligent technology and equipment for liquor picking.

In 2012, a liquor picking technology was developed based on a distributed control system (DCS) [50,51]. An online detector was set in a pipe to detect the contents of ethanol and aroma compounds in liquor, based on an electric signal. The DCS was used to control electromagnetic valves based on a comparison of the real-time detection data with default data. According to the above design, automatic liquor picking was achieved.

Similar to the above design, Yu et al. [52] designed an automatic technology based on a PLC controller for liquor picking in 2019. The ethanol content was monitored online, based on changes in the dielectric constant in the distillate. The central control system controlled the switching of multiple electromagnetic valves to collect different base Baijiu from different distillation stages.

Current applications are mainly based on differences in the physical parameters of liquors with different ethanol contents, such as differences in density. For example, Shilixiang Co., Ltd. (Botou, China) [53] reported an intelligent liquor picking robot that obtained the temperature and density from an online detection device; this information could be converted to the ethanol content of the liquor. The static and dynamic accuracies of the measurements were ±0.2% vol and ±0.5% vol, respectively. The display accuracy was ±0.5% vol. This type of measurement method is based on the static differential pressure, which is only slightly influenced by the fluctuation of the liquid. The computer system controls the switching of the valves according to the signals from the online detector to achieve intelligent liquor picking.

In addition, differences in the molecular structure can result in a difference in the spectrum. Thus, near-infrared spectroscopy [54] and Raman and fluorescence spectroscopy [55] can be used for online detection of the ethanol content in liquor. Furthermore, spectral analysis technology is expected to achieve a rapid detection of the aroma compounds in liquor, to thereby rank the distillates of the liquor. In addition, automatic liquor-picking can be achieved by collecting images of the “Jiuhua” during the distillation process [56]. However, owing to the complexity and fluctuation of the “Jiuhua”, the collection of “Jiuhua” images requires specific equipment and conditions, such as a lens with anti-fog performance and a good light source.

The online detector for alcohol content is effective and has already been applied in some distilleries. However, this method ignores the flavor components, and a more accurate ranking is still needed. Further attention should be paid to the online picking and ranking of liquor according to both concentrations of alcohol and flavor components, which is an important and challenging direction for intelligent liquor-picking.

## 5. Storage and Blending

The base Baijiu from different stages of the distillation process is usually stored for a period of time (usually 1–3 years), and a blending process is also required before the liquors leave for filling and packaging. In the past, pottery jars and Jiuhai (a large Baijiu container) have mainly been used for the storage stage. At present, stainless steel tanks are widely used for the storage of common Baijiu or for the temporary storage of liquor for convenient filling; in contrast, pottery jars are mainly used for the storage of high-quality Baijiu. When using stainless steel tanks for Baijiu storage, the sensors of temperature and liquid level can be installed. Moreover, cooling spray devices and liquid level alarms can also be added to achieve automatic and intelligent management during storage [57]. Pret Information Technology Co., Ltd. (Taian, Shandong) launched a digital tank metering system [58] that not only monitored the parameters of each storage tank (including the liquid level, temperature, and density) in real time but also dynamically displayed them on the screen. Furthermore, the data could be remotely monitored and managed through the network. The intelligent monitoring system for storage is similar to that in the digital starter-making workshop mentioned above, although in addition to monitoring the parameters, fire safety should be paid more attention. If the storage time of the Baijiu reaches the expected time, the Baijiu proceeds to the last step before leaving the factory, i.e., blending.

The purpose of blending is to eliminate the flavor differences among different batches of Baijiu and to coordinate and balance the aroma components through “blending liquor by liquor”. Blending is of great importance for ensuring that the final products have a unified quality. Additionally, blending is also an important part of the development of a new flavor-type of liquor. The blending process mainly comprises two sections: (1) a sensory evaluation of small tested samples to determine the appropriate blending proportions of different base liquors and water, and (2) an amplification experiment, based on using a large-scale container in the production workshop according to the optimized blending proportion. The former urgently requires improvements in efficiency from intelligent technology owing to the excessive time required, whereas the latter also urgently needs to realize automation to increase the success rate of the amplification experiments. Compared with Baijiu brewing processes, studies on automatic and intelligent blending started relatively early. In the early 1980s, Wuliangye Group Co., Ltd. (Yibin, China) developed computer blending technology [59]. At present, this technology has been developed into integrated technologies, including those for liquor storage, physical and chemical analyses of liquor, management of liquor stores, and blending. In the 21st century, with the further developments in automation technology, automatic blending systems have been successively applied in many distilleries [60,61,62]. These automation systems can be applied for large-scale blending in workshops to increase the accuracy of blending and avoid repeated blending operations. For example, Hubei Baiyunbian Liquor Industry Co., Ltd. (Jingzhou, China), a typical representative of Jian-flavor Baijiu, has a unique and complicated blending process [63]; there are 221 tanks in the blending tank zone. Automatic blending control is achieved based on the interconnections of pipelines and electric valves between the tanks. Base Baijiu or soft water can be automatically fed to the tank from the pipeline. During the blending, the operators only need to select and set the necessary parameters for correct blending. After the system starts, the liquor and water are transported to the corresponding tanks according to the tank numbers and required amounts of liquor or water for each tank. The valve and pump are shut off after the pre-set capacity of one tank is reached. Then, the valve of the next tank is opened automatically for the following process until the formula is completed. “One-click blending” has basically been achieved by using this automatic control system. As a result, the quality of the Baijiu product is stable; moreover, the system provides a doubled brewing capacity and comprehensive production capacity. The automatic blending system promotes the overall improvement of the economic and social benefits of enterprises.

Relatively few studies have been conducted on the blending of small liquor samples. This is mainly owing to the fact that this process strongly relies on the sensory evaluation of Baijiu tasters so much that the need for mechanization and automation is not urgent. However, the number of manual blended samples can potentially be large, especially when developing a new product. In such cases, the workload for manual blending and sensory evaluators is actually very heavy. To reduce the intensity of manual blending work, an intelligent blending system for small samples that combined humans and machines was established [64]. First, according to the physicochemical properties and sensory characteristics of standard samples in a commercial Baijiu database, the targeted flavor type of the liquor was selected. Second, the blending method of the required flavor-type was obtained from the blending knowledge of the Baijiu selection database. This database includes data on the blending methods for former standard commercial liquors and the physicochemical properties and sensory characteristics of base liquors, as well as other related knowledge, rules, and experiences. Third, the formula was designed based on an expert system for blending. Finally, the formula was further modified using a human–machine interface, after sensory evaluations by Baijiu tasters.

Current studies on automatic and intelligent blending have focused on the optimization of an automatic control system for liquor blending in large containers [44,65], and on the application of control algorithms [66]. However, there are relatively few studies on small automatic and intelligent blending systems for Baijiu tasters, although automatic mixers have appeared in other alcoholic beverage industries. Nonetheless, the automation of blending has been achieved the earliest and is reported the most among all of the above-mentioned procedures for Baijiu production; this is closely related to the clarity regarding the influencing factors for blending.

## 6. Automatic Production Line for Baijiu

A mechanized production line has been built in most manufacturing plants for Baijiu, but the continuity between different production parts is generally low [41,67]. Recently, continuous and automatic production lines for SSF Baijiu have been gradually built for MFB production after the continuous operations for starter-making, fermentation, and distillation have been achieved.

Shandong Jingzhi Liquor Co., Ltd. (Weifang, China) [68] built 13 automatic and intelligent production lines for NFB. Eight sets of key equipment for intelligent and automatic starter addition, steamer-filing, and air cooling were applied in each production line. In addition, remote monitoring was achieved using the Internet. Quantitative control was achieved for the ratio of raw grains to fermented grains, amounts of bran and Jiuqu, and physical and chemical properties of the grains before being placed into the pit. As a result, the operation process was stable, and the process execution rate was more than 98%. Owing to the implementation of automatic and intelligent production for NFB, the average liquor yield was 3% higher than with manual operation.

Relatively speaking, the degree of automation and intelligence for the Xiaoqu MFB produced by SSF is higher because of the use of the SSFT. For example, in 2010, Fenglin distillery of Jingpai Co., Ltd. (Huangshi, China) used an SSFT as the fermentation container for the brewing of Xiaoqu MFB for the first time [69]. Then, a complete automatic production line including grain steaming, cooling, Jiuqu addition, saccharification, fermentation, and distillation was gradually established as the “digital intelligent brewing demonstration workshop of Chinese Baijiu”. The production process is shown in Figure 9, and it is described as follows [70]. First, the raw grains were soaked in a stainless-steel tank for 20–24 h; then, they were transported by self-weight to a rotary grain steaming pot for high-pressure cooking. After cooking, the steamed grains were cooled down, and the Jiuqu was automatically added and mixed with the steamed grains. Then, the mixture was transported by a chain plate conveyor belt at a slow speed to saccharification workshops equipped with temperature-controlling equipment as shown in Figure 10a. After saccharification for 24 h, screw conveyors were used to evenly mix the saccharified and distilled grains, and movable fermentation tanks were used as the fermentation containers, as shown in Figure 10b. The fermentation tanks were placed in a fermentation chamber with a temperature control system for 15–18 days. After fermentation, the tanks with the fermented grains were placed on turnover machines by the forklift trucks, so as to discharge the grains to be transported to the distillation part by conveyor belts. The steamer-filling was conducted by intelligent robots, and automatic liquor picking was realized based on an online monitoring of the alcohol content. The obtained base liquor was weighed on site and was transported to the storage tanks directly through a pipeline. The distilled grains were discharged into another container through an openable bottom of the steamer. During the entire production process, only one or two workers were required to inspect the production line in the workshop as shown in Figure 10c; another few workers monitored the production parameters in the control center. The number of workers in the brewing workshop was reduced by 80% after the above production line was used, whereas the production efficiency was greatly improved.

The main reasons for realizing the complete automation of the production line of SSF for Xiaoqu MFB are as follows: (1) there is no need to move the materials in and out of the pit layer by layer, (2) the fermentation does not need the pit mud, and (3) the effects of the automation production on the Baijiu quality are small.

## 7. Conclusions and Prospects

With the rapid development of information technology, countries worldwide have proposed important strategic plans for the development of their manufacturing industries. For example, China’s “Made in China 2025” and Germany’s “Industry 4.0” aim to promote the automatic and intelligent upgrading of the manufacturing industry, which is crucial for increasing production efficiency and promoting industrial transformation. The production of Chinese Baijiu adopts a unique solid-state brewing process, which is quite different from the liquid-state brewing processes for other distilled spirits. Thus, the automatic production equipment for liquid-state brewing is not completely applicable to the production of Chinese Baijiu, especially for SSF and SSD processes. The relevant automatic equipment can only be developed independently in China, so as to improve production efficiency and obtain stable product quality. Generally speaking, the Baijiu manufacturing industry presents the wide popularization of mechanization, continuous development of automation, and budding of intellectualization. An “intelligent brewing” workshop based on information technology (such as Internet of Things, cloud computing, big data analysis, and intelligent robots) has been applied in Baijiu production. Currently, the following problems and challenges remain concerning the upgrading and development of the Baijiu industry.

(1) Control of the SSF process: Owing to the limited understanding of the SSF mechanisms, it is difficult to transfer pits for other flavor types of Baijiu, especially for NFB and JFB, from underground to overground. As a result, the manual work is necessary for the relevant operation for fermentation of these liquors, and the monitoring of the fermentation parameters is also difficult. As for the control of SSF processes of Baijiu, that is even more difficult. In other words, fermentation is the most difficult process insofar as realizing automatic and intelligent upgrading for Baijiu production. In the future, research on the mechanisms of SSF, especially the metabolic mechanisms of the microbial community, should be strengthened to develop artificial pit muds and novel fermentation tanks, or to study fermentation technologies without mud for NFB or JFB.

(2) Accuracy of the digital sensors: At present, the construction of the digital workshop for Qu-making and the intelligent liquor picking process strongly depend on the efficient and accurate responses of the sensors. The working efficiency of the sensors is closely related to their working environment. For example, pollution of the probe at the sensing end, or changes in the pH and/or flow type may affect the sensing accuracy. However, at present, there is a lack of comprehensive and systematic studies on the applicability, efficiency, and maintenance of sensors in different environments.

(3) Security of intelligent monitoring on the Internet: Owing to the rapid development of the Internet of Things, the parameters of the Baijiu production workshop can be transmitted to a cloud as big data for people to monitor, judge, and make decisions. These data are the most confidential production information for a manufacturer, and should be transmitted with higher security requirements; otherwise, there is high risk of the leakage of technical secrets. Thus, the data security problem cannot be ignored in network control.

(4) Intelligent ranking of Baijiu: The online technology for the accurate cutting of liquor distillates according to the contents and types of aromatic substances (instead of just ethanol content) must be further developed, which will save manpower and greatly increase the production efficiency. The key lies in the rapid identification of the different trace amounts of aromatic substances. The establishment of a quantitative relationship between the aromatic components and sensory evaluations (from tasters) is a very challenging research direction.

(5) Costs for upgrading technology and equipment in Baijiu manufacturing: At present, only large-scale Baijiu plants can afford transformations of digital and intelligent workshops and equipment. Small-scale Baijiu enterprises are limited by costs and cannot accept an automatic production line. Therefore, the cost of intelligent equipment (such as steamer-filling robots) is expected to be reduced to meet acceptable limits for such Baijiu manufacturers.

(6) Automatic and intelligent production changing the original flavor of the liquor and affecting consumer acceptance: This is something that many producers are very worried about, although the liquor flavors produced by a modern production line are very close to the corresponding traditional flavors. Nevertheless, modern production lines must be further improved to enhance the flavor of Baijiu; simultaneously, producers should put aside their misgivings and have faith in modern production techniques.

At present, the development directions for the liquor industry are oriented toward the modernization of production, internationalization of the market, and improving the health of products under the dual guidance of flavor and health. The industrial upgrading of Baijiu production represents the general trend for brewers worldwide. Intelligent equipment is being developed, such as robot inspectors for liquor storehouses and workshop displays via virtual reality technology. The upgrading of the Baijiu industry will usher in an era of “intelligent brewing” for Baijiu and provide a good reference for other brewing industries around the world.

## Figures and Tables

**Figure 1 foods-10-00680-f001:**
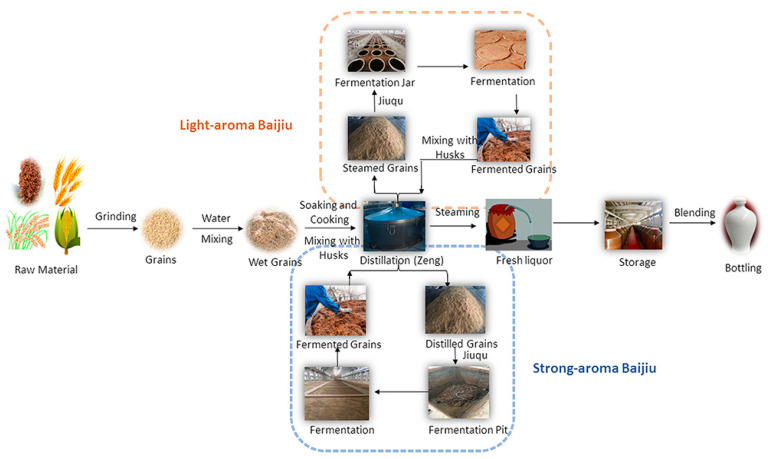
Traditional production processing of Baijiu (Image reprinted from ref. [3], Copyright (2018), with permission from American Chemical Society).

**Figure 2 foods-10-00680-f002:**
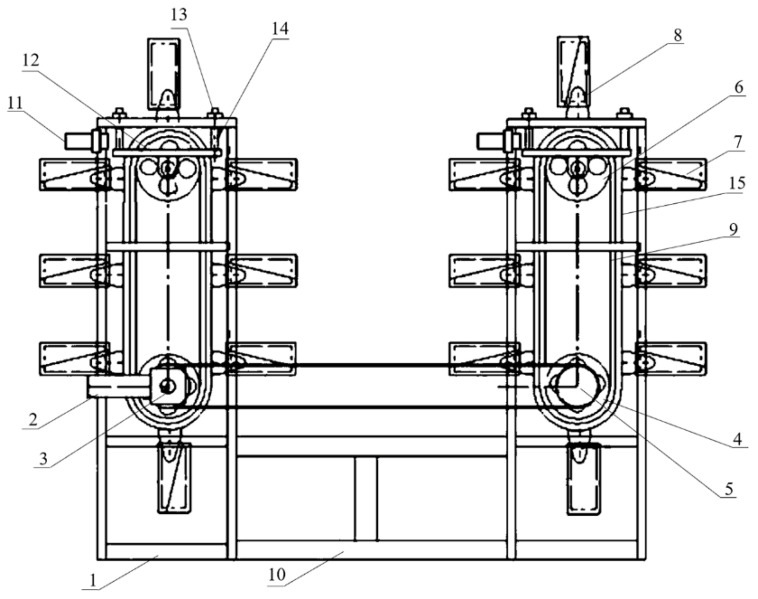
Automatic device for the operation to turn over the Daqu (1: rack of device for turning over Daqu; 2: power component; 3: transmission component; 4: driving wheel; 5: transmission wheel; 6: driven wheel; 7: box for loading Daqu; 8: connecting bracket; 9: driving chain/belt; 10: connecting rod; 11: proximity sensor; 12: tensioning bracket; 13: tensioning nut; 14: tensioning screw; 15: slide rail). Image reprinted from ref. [11], Copyright 2021, with permission from Anhui Gujing Distillery Co. Ltd. (Bozhou, China).

**Figure 3 foods-10-00680-f003:**
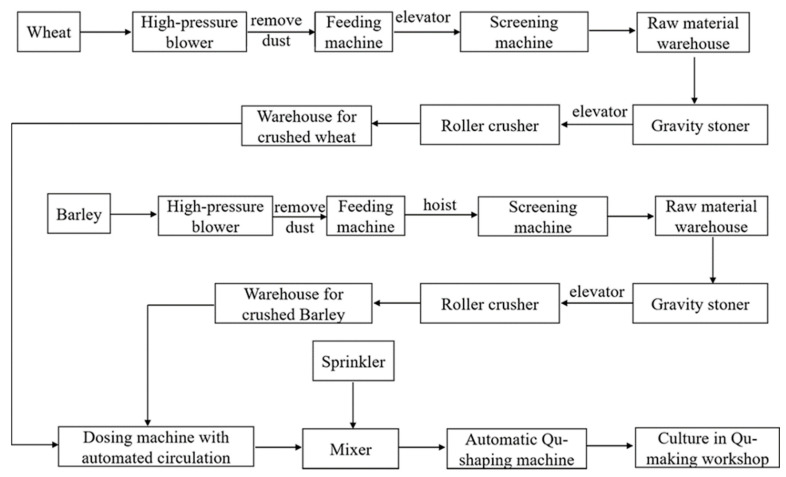
Intelligent Koji-making system for the production of Daqu for Nong-flavor Baijiu, adapted from reference [13].

**Figure 4 foods-10-00680-f004:**
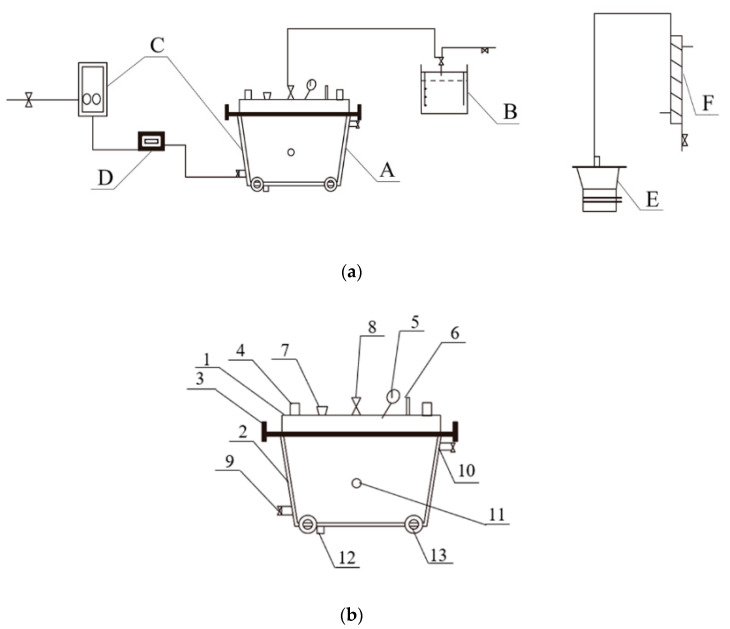
Schematic diagram of the automatic brewing system and artificial pit. (**a**) The moveable automatic brewing system (A, artificial pit; B, gas-collection equipment; C, heating and cooling equipment using circulating water; D, temperature control equipment; E, distilling still for stewing grains and Zaopei; F, receiving device for alcohol condensation.); (**b**) the artificial pit [1, pit cover; 2, pit body (including an interlayer); 3, hold-down bolt; 4, handle; 5, manometer; 6, thermometer; 7, sampling filler; 8, bifurcated vent; 9, infall; 10, outlet; 11, temperature probe; 12, vent for sampling and draining; 13, pulley] (Image reprinted from ref. [23], Copyright 2021, with permission from John Wiley and Sons).

**Figure 5 foods-10-00680-f005:**
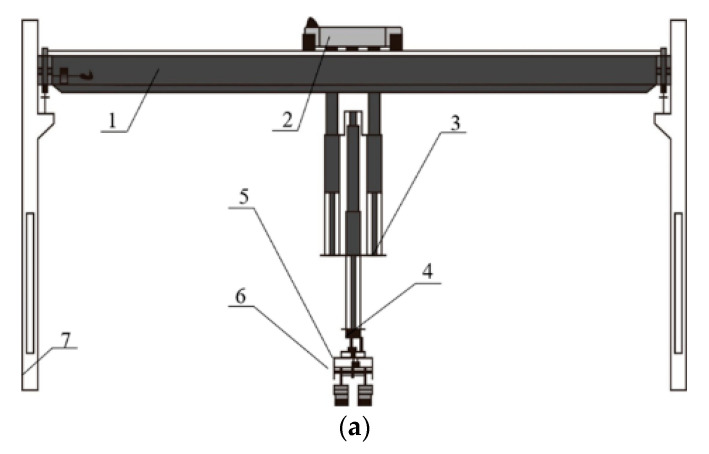
Device for discharging the fermented grains. (**a**) Continuous grabbing device (1, crossbeam; 2, car; 3, elevator component; 4, quick-changing component; 5, rotation component; 6, grains-grabbing device; 7, wall beam); (**b**) continuous lifting device (1, horizontal guide rail; 2, longitudinal guide rail; 3, main rack; 3-1, hydraulic device; 3-2, electric motor; 4, secondary rack; 4-1, roller and fastening device; 4-2, electric motor; 5, device for lifting fermented grains; 5-1, device for scooping fermented grains; 5-2, conveyor; 6, handcart; 7, pit). Image reprinted from ref. [30] and [31], Copyright 2021, with permission from authors.

**Figure 6 foods-10-00680-f006:**
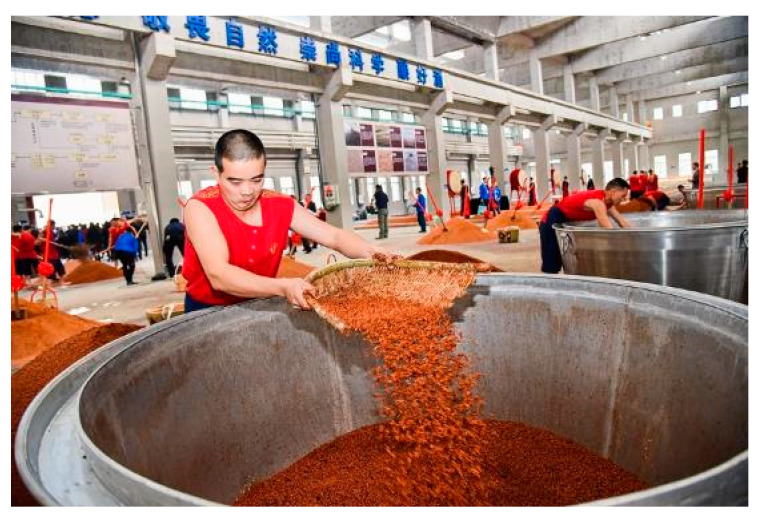
Traditional manual steamer-filling (kindly provided by Sichuan Langjiu Group Co., Ltd., Luzhou, China).

**Figure 7 foods-10-00680-f007:**
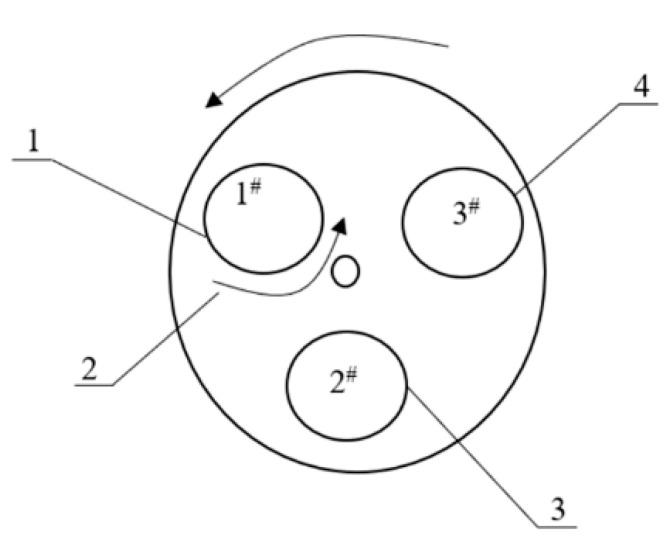
Rotary table Zeng with three working stations (1: steamer-filling station; 2: rotation of Zeng; 3: distillation station; 4: discharging station for distilled grains). Image reprinted from ref. [39], Copyright 2021, with permission from authors.

**Figure 8 foods-10-00680-f008:**
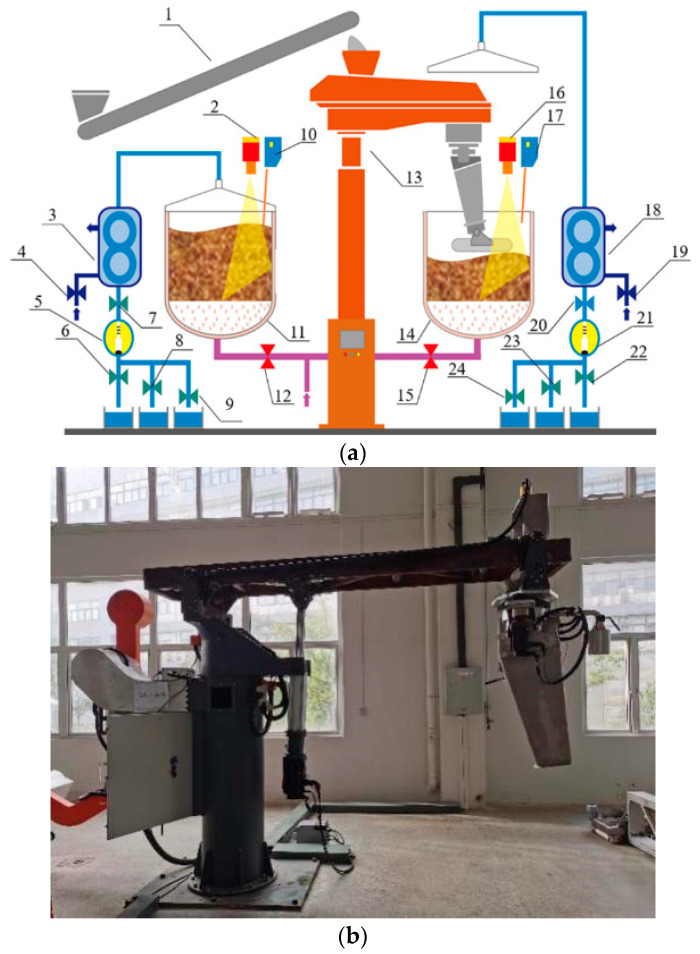
Robot for steamer-filling. (**a**) Design of intelligent distillation system using a robot for steamer-filling (1, conveyor; 2 and 16, temperature detector; 3 and 18, condenser; 4 and 19, valve for cooling water; 5 and 21, detector of alcohol content; 6 and 22, valve for foreshot of liquor; 7 and 20, main valve for distillate; 8 and 23, valve for liquor body; 9 and 24, valve for tail liquor; 10 and 17, detector of grain surface; 11 and 14, Zeng; 12 and 15, steam valve; 13, steamer-filling robot); (**b**) Digital figure of robot for steamer-filling (the figures are all kindly supplied by Wuhan Fenjin Intelligent Machine Co., Ltd., Wuhan, China).

**Figure 9 foods-10-00680-f009:**
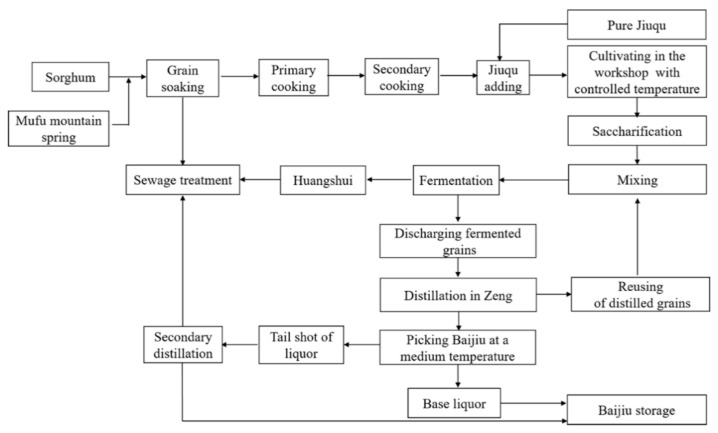
Automatic brewing process of solid-state fermentation for Xiaoqu light flavor Baijiu (kindly provided by Fenglin distillery of Jingpai Co., Ltd., Huangshi, China).

**Figure 10 foods-10-00680-f010:**
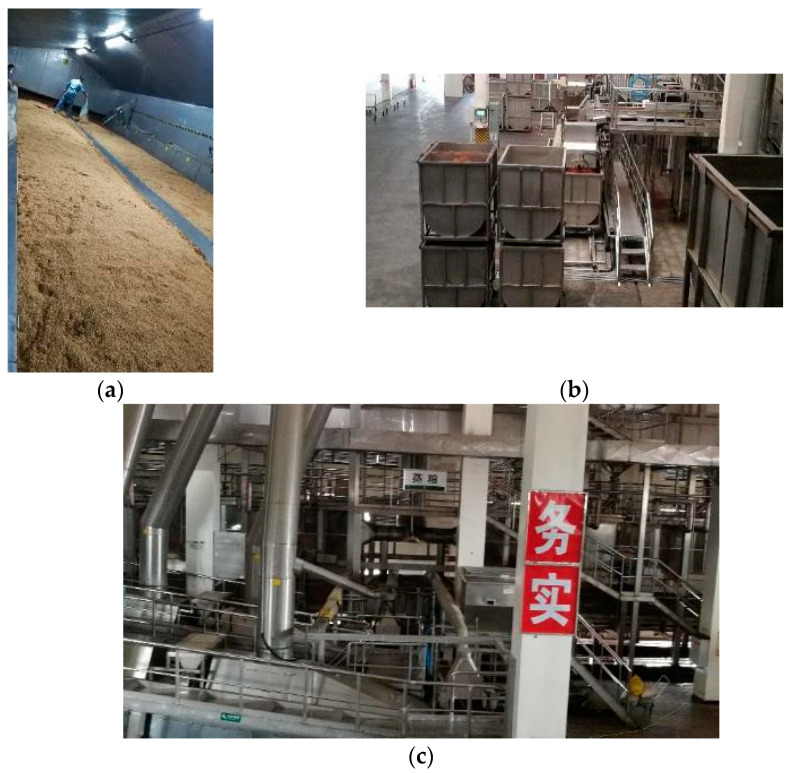
Automatic production line for Xiaoqu mild flavor Baijiu. (**a**) Saccharification workshop; (**b**) fermentation tanks; (**c**) production workshop (the figures are kindly provided by Fenglin distillery of Jingpai Co., Ltd., Huangshi, China).

**Table 1 foods-10-00680-t001:** Effect of automatic steamer-filling on the quality of Baijiu.

Methods of Steamer-Filling and Flavor Styles	Improvement of Production Efficiencies (Compared with Manual Steamer-Filling)								Ref.
	Liquor Yields	Rate of High-Quality Liquor	Total Acids	Total Esters	Ethyl Hexanoate	Ethyl Lactate	Sensory Comparison	Time and Manpower	
Screw conveyors + rotary Zeng, Nong-flavor Baijiu	Unchanged	Increased	Unchanged	Decreased	Increased	Decreased	Reduced pungency	The times for steamer filling, distillation, and discharging were overlapped, and the efficiency was improved.	[39]
LabVIEW + industrial robot	Increased by 11.4%	Increased by 10.5%							[43]
Automatic steamer-filling with inverted propeller, Nong-flavor Baijiu	Ratio of Baijiu body was increased by 1.7%	Increased by 26.8%	Unchanged	Increased by 1.03 g/L	Increased by 0.51 g/L	Decreased	Unique soft, sweet, and clean sensory characteristics	The time was shortened by 12 min.	[46]
Rotary feeding machine, Nong-flavor and sesame flavor Baijiu	Decreased by 0.1 kg	Increased by 12%			Increased by 10.8%	Decreased			[41]
Steamer-filling robot with infrared temperature sensor, Nong-flavor Baijiu	Unchanged	Increased by 11.33%	The main aroma components achieved an increase of 16%.					The number of workers in a group was reduced from seven to four.	[47]
Steamer-filling robot with infrared temperature sensor, Jiang-flavor Baijiu	Increased by 6.5%	-	-				No significant difference	The time required for steamer-filling was shortened, and the number of workers was reduced from five to three.	[48]

## Data Availability

Data sharing not applicable. No new data were created or analyzed in this study.

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
