# Peer review of "Automatic and Intelligent Technologies of Solid-State Fermentation Process of Baijiu Production: Applications, Challenges, and Prospects"

_foods, 2021, doi:10.3390/foods10030680_

Round 1
Reviewer 1 Report
The review “ Recent advances in the automatic and intelligent technologies of solid-state fermentation process of Baijiu production: applications, challenges and prospects” thoroughly investigates the problems correlating to the automatic and intelligent production processes of Baijiu with repercussions in terms of costs and human labor.
Moreover, there are some points that need to be clarified or corrected:
- The title does not fully reflect the content of the paper. In my opinion, “recent advances” should be eliminated as it seems to be only the state of the art and many points many points are still not resolved. So, I suggest this tile: “Automatic and intelligent technologies of solid-state fermentation process of Baijiu production: applications, challenges and prospects”.
- Line 47: the authors should specify the “appropriate temperature” to cool the mixture in a solid state.
- In the section 2, it would be interesting to insert the microbiological composition of Daqu, Xiaoqu and Fuqu. In particular, at line 89 what is the pure mold using for artificial inoculation of Fuqu?
- Line 129: what does “apparent morphology” of Daqu mean? And can the authors explain how the observation of this morphology is carried out?
- Line 160: if it possible, the authors should insert the data relating to the microbial diversity indicators and dominant microbial community in the intelligently made Koji in comparison to those in traditional artificial Koji.
- Change the title of the subsection 3.1 in this way: “Monitoring of fermentation temperature” because in this section there is no mention of other parameters.
- Line 290-291: the authors should insert the data relating to the liquor yields and the contents of total acids and total esters for the liquors obtained in the stainless steel and pottery jars, respectively.
- Line 308-311: the authors should insert the data relating to the contents of the total ester, acetic acid, and ethyl lactate in the Baijiu produced in the overground stainless-steel vessel and in the underground stainless-steel vessel.
- I suggest to delete the section 7 “Comparison of current development statuses of major brewing industries” that is not related with the review because the topic is focused on Baijiu production.
Reviewer 2 Report
The authors presented the traditional and modern method of Baijiu production. The manuscript introduces the reader to this traditional Chinese alcoholic drink. However, solid-state fermentation technology is not new and has already been described many times. Therefore, the work does not represent a high scientific level. However, it is interesting from a cognitive point of view.
Minor corrections:
Line 33-34 “Baijiu is the world's most-consumed spirit, for example, the gross sales of Baijiu in 2019 were more than $ 80 billion” - This statement needs to be revised - Baijiu is not very popular around the world and consumption of other beverages in Europe and America is much higher.
Page numbering requires correction - from page 16.
The work contains punctuation errors - please correct (e.g. line 590)
Reviewer 3 Report
The present manuscript is intended to review the recent advances in using intelligence technology and automatic devices for the production of Baijiu in solid state fermentation. In general, the manuscript is well motivated and structured, highlighting the main points that can be automated in each process steps and having a critical discussion the most important bottlenecks that must be overcome. The different sections are well connected between each other and authors provide at the end of the manuscript few lines describing what are the major needs of this industry in the short term.
